# Characterization of Key Compounds of Organic Acids and Aroma Volatiles in Fruits of Different *Actinidia argute* Resources Based on High-Performance Liquid Chromatography (HPLC) and Headspace Gas Chromatography–Ion Mobility Spectrometry (HS-GC-IMS)

**DOI:** 10.3390/foods12193615

**Published:** 2023-09-28

**Authors:** Yanli He, Hongyan Qin, Jinli Wen, Weiyu Cao, Yiping Yan, Yining Sun, Pengqiang Yuan, Bowei Sun, Shutian Fan, Wenpeng Lu, Changyu Li

**Affiliations:** 1Institute of Special Animal and Plant Sciences, Chinese Academy of Agricultural Sciences, Changchun 130112, China; 82101215184@caas.cn (Y.H.); qinhongyan@caas.cn (H.Q.); 82101215188@caas.cn (J.W.); 82101202231@caas.cn (W.C.); 82101225211@caas.cn (Y.Y.); 82101225210@caas.cn (Y.S.); 82101222242@caas.cn (P.Y.); fanshutian@caas.cn (S.F.); luwenpeng@caas.cn (W.L.); 2Faculty of Agriculture, Yanbian University, Yanji 136200, China; 2022050841@ybu.edu.cn

**Keywords:** *Actinidia argute* resources, organic acid, volatile compound, orthogonal partial least squares discriminant analysis, odor activity value

## Abstract

*Actinidia arguta*, known for its distinctive flavor and high nutritional value, has seen an increase in cultivation and variety identification. However, the characterization of its volatile aroma compounds remains limited. This study aimed to understand the flavor quality and key volatile aroma compounds of different *A. arguta* fruits. We examined 35 *A. arguta* resource fruits for soluble sugars, titratable acids, and sugar–acid ratios. Their organic acids and volatile aroma compounds were analyzed using high-performance liquid chromatography (HPLC) and headspace gas chromatography–ion mobility spectrometry (HS-GC-IMS). The study found that among the 35 samples tested, S12 had a higher sugar–acid ratio due to its higher sugar content despite having a high titratable acid content, making its fruit flavor superior to other sources. The *A. arguta* resource fruits can be classified into two types: those dominated by citric acid and those dominated by quinic acid. The analysis identified a total of 76 volatile aroma substances in 35 *A. arguta* resource fruits. These included 18 esters, 14 alcohols, 16 ketones, 12 aldehydes, seven terpenes, three pyrazines, two furans, two acids, and two other compounds. Aldehydes had the highest relative content of total volatile compounds. Using the orthogonal partial least squares discriminant method (OPLS-DA) analysis, with the 76 volatile aroma substances as dependent variables and different soft date kiwifruit resources as independent variables, 33 volatile aroma substances with variable importance in projection (VIP) greater than 1 were identified as the main aroma substances of *A. arguta* resource fruits. The volatile aroma compounds with VIP values greater than 1 were analyzed for odor activity value (OAV). The OAV values of isoamyl acetate, 3-methyl-1-butanol, 1-hexanol, and butanal were significantly higher than those of the other compounds. This suggests that these four volatile compounds contribute more to the overall aroma of *A. arguta*. This study is significant for understanding the differences between the fruit aromas of different *A. arguta* resources and for scientifically recognizing the characteristic compounds of the fruit aromas of different *A. arguta* resources.

## 1. Introduction

*Actinidia arguta* [(Sieb. & Zucc) Planch. ex Miq.], also known as soft dates, kiwi berries, kiwi pears, and more, is a large deciduous liana from the kiwifruit family (Actinidiaceae Gilg & Werderm.) and the kiwifruit genus (Actinidia Lindl) [1]. This characteristic berry resource is native to China, with wild resources also found in Japan, the Korean Peninsula, and the Russian Far East [2,3]. Its fruits are tasty and unique in flavor and rich in nutrients, such as proteins, vitamins, amino acids, minerals, dietary fiber [4], polysaccharides, polyphenols, alkaloids, volatile oils, proanthocyanidins, and other active ingredients [5], which have antitumor, antiradiation, antioxidant, antiaging, hypoglycemic, anti-inflammatory, insomnia-inhibiting, immunity-improving, and laxative functions [6,7,8,9]. Nowadays, *A. arguta* is popular with the public and the market for its rich nutritional and medicinal value.

Volatile aroma substances are crucial factors that influence fruit quality and consumer enjoyment [10] as well as important indicators of fruit flavor quality. Research on the various aromas of fruits can provide a theoretical basis for screening superior resources and help to better understand and control key flavor quality parameters that may affect fruit processing [11]. Fruit volatile aroma substances are influenced by various factors, such as variety, cultivation conditions, climatic conditions, ripening period, and storage conditions [12,13]. Dozens of compounds, mainly esters, alcohols, aldehydes, alkenes, and ketones, have been identified in the fruits of *A. arguta* varieties [14,15]. However, previous studies on volatile aroma substances of *A. arguta* have mainly focused on varieties and wine products [14,16]. Sun Yang et al. [15] detected 41 compounds from the fruits of different *A. arguta* varieties. There were differences in the types and contents of the aroma components between varieties, with ‘Autumn Honey’ having the highest number of the kinds of aroma substances. Zhang Baoxiang et al. [17] detected 56 aroma substances in different varieties of *A. arguta*-brewed dry wine, clarified the composition and content of 46 of them, and found that the aroma components of different types of brewed dry wine were the same, but the range varied greatly through analysis. Little research has been performed on the volatile aroma substances of *A. arguta* resource fruits, which should be considered. Meanwhile, the differences between the volatile aroma components of different *A. arguta* resource fruits are not apparent. Therefore, this study aimed to detect their volatile aroma components and to identify the main compounds that affect the volatile aroma components of *A. arguta* resource fruits.

Currently, the commonly used methods for the detection and analysis of fruit aroma substances are gas chromatography–mass spectrometry (GC-MS), gas chromatography–ion mobility chromatography (GC-IMS), and gas chromatography–olfactometry (GC-O-MS) [17,18,19]. However, GC-MS and GC-O-MS have several disadvantages, including the need for sample pre-treatment, a more complex operation process, a long assay time, and excessive sample consumption [20]. The pre-treatment process may cause damage to the aroma substances present in the models themselves, leading to differences in the types and contents of the detected aroma substances [21]. On the other hand, GC-IMS is an instrumental analytical technique that separates ions of the detected substance according to their ion mobility at atmospheric pressure. It has several advantages, such as simple sample preparation, easy operation, high sensitivity, fast analytical speed, and even trace amounts of volatile compounds can be detected [22,23,24]. In addition, ion mobility can significantly separate isomers and isobaric compounds [25]. GC-IMS is a recently discovered analytical technique for detecting volatile compounds in mixed analytes [26]. It combines the separation properties of GC with the fast correspondence and high sensitivity of IMS, which allows the detection of alcohols, esters, aldehydes, ketones, and aromatics, including even the most complex and problematic matrices [27], and has been widely used for the study of volatile compounds in food sciences, e.g., in kiwifruit [19], jujube [28], melons [29], wines [30], eggs [31], and honey [32]. Compared with GC-MS, GC-IMS does not require sample pre-processing and preserves the original aroma components of the sample intact. Multivariate statistical methods, such as principal component analysis (PCA) modeling, orthogonal partial least squares discriminant analysis (OPLS-DA) modeling, and cluster analysis, are commonly used when analyzing GC-IMS volatiles. Principal component analysis (PCA) is based on the principle of KL transformation. It uses the idea of dimensionality reduction to transform multiple indicators into a small number of major components that can reflect most of the information of the original variables [33]. Orthogonal partial least squares discriminant analysis (OPLS-DA) is a supervised statistical method of discriminant analysis, and PCA-based OPLS-DA further inputs the transformed score information into the model, identifying the key contributors to the variance-related variables in the model [34,35]. Hierarchical cluster analysis (HCA) calculates the correlation between samples using defined criteria, which are simplified and combined according to the degree of correlation to provide a more intuitive and comprehensive comparison of similar varieties and components [36]. Therefore, HS-GC-IMS mixed multivariate statistical methods have been widely used in metabolomics and flavoromics studies [37,38].

In this study, the sugar and acid contents of 35 *A. arguta* resource fruits were determined. The volatile aroma components were rapidly analyzed and detected by HS-GC-IMS technology. This produced a top view of the differences and established the fingerprints of volatile aroma compounds of different *A. arguta* resource fruits. Furthermore, based on volatile aroma compounds, a quantitative descriptive analysis of the data was performed through multivariate statistical analysis to analyze the differences in volatile aroma compounds between individual resources. In addition, principal component analysis, OPLS-DA analysis, and OAV analysis were combined to screen essential volatile compounds affecting the fruit flavor of *A. arguta* resources. This study provides a theoretical basis for screening *A. arguta* resources with excellent flavor quality, enhancing and improving the flavor quality of *A. arguta* processed products. It also aids in scientifically recognizing the characteristic compounds of the fruit aroma of different *A. arguta* resources and provides a theoretical basis for regulating the flavor quality of processed products.

## 2. Materials and Methods

### 2.1. Materials and Reagents

#### 2.1.1. Materials

The 35 resources selected for this study (Table 1) were sampled from the *Actinidia arguta* Resource Nursery of the Institute of Special Animal and Plant Sciences of the Chinese Academy of Agricultural Sciences, Zuojia Town, Jilin City, Jilin Province, China (44°00′ N; 126°01′ E). The sampling time was September 2022, when the fruits were ripe. Sampling was performed by randomly selecting well-grown, medium-sized vines in the resource nursery, choosing soft date palm kiwifruit with the same degree of exposure to light, the same size, and similar hardness and fruit that was free of pests and diseases. We picked about 300 g of fruit from each resource, placed the samples in separate numbered sampling bags, and transported them back to the lab in an insulated box. We placed the fruit in a −80 °C refrigerator for storage after measuring the relevant indicators on the same day.

#### 2.1.2. Reagents

Analytical purity: anthrone (Sinopharm Chemical Reagent Co., Ltd. Shanghai, China); ethyl acetate, concentrated sulfuric acid, phosphoric acid (Beijing Chemical Factory, Beijing, China).

Chromatographic purity: methanol (TEDIA reagent, Fairfield, OH, USA); oxalic acid, quinic acid, malic acid, shikimic acid, lactic acid, citric acid, ascorbic acid (Shanghai Yuanye Biotechnology Co., Ltd. Shanghai, China); 4-methyl-2-pentanol (Shanghai Lianshuo Biotechnology Co., Ltd. Shanghai, China).

### 2.2. Instruments and Equipment

High-performance liquid chromatograph (Agilent Technologies, Waldbronn, Germany); FlavourSpec^®^ Flavour Analyzer (G.A.S. It is based on gas chromatography ion mobility spectrometry (GC-IMS), which has both the high separation of gas chromatography and the high sensitivity of ion mobility spectrometry, and can detect trace volatile organic compounds in the samples without enrichment and concentration and other pre-processing to maintain the original flavor of the flavor samples, which is very suitable for the analysis of aroma components. The accompanying software can generate the sample aroma fingerprints, which can easily realize the comparison of sample differences and consistency control); CJJ-931 dual-magnetic heating stirrer (Jiangsu Jintan Jincheng Guosheng Experimental Instrument Factory, Jiangsu); hgs-12 electric thermostatic water bath, KQ-300E ultrasonic cleaner snowflake ice machine (Beijing Changliu Scientific Instrument Co., Ltd. Beijing, China); FA1004B electronic balance (Shanghai Yue Ping Scientific Instrument Co., Ltd. Shanghai, China); IMark enzyme labeling instrument (Biorad, Philadelphia, PA, USA); high-speed freezing centrifuge (Allegra 64R, USA); −80 °C ultra-low-temperature refrigerator (Beijing Chengmaoxing Science and Technology Development Co., Ltd. Beijing, China); WAX columns (RESTEK, Bellefonte, PA, USA).

### 2.3. Methods

#### 2.3.1. Determination of Soluble Sugar and Titratable Acid Content

Soluble sugar content was determined by the anthrone reagent method, and titratable acid content was determined by titration method with sodium hydroxide solution, both referring to the *Experiment Guideline of Postharvest Physiology and Biochemistry of Fruits and Vegetables* (1st edition, December 2020). Sugar–acid ratio = soluble sugar content/titratable acid content.

#### 2.3.2. Determination of Organic Acid Content

The organic acid content was determined by high-performance liquid chromatography (HPLC), referring to the previously published literature [39]. Oxalic acid, quinic acid, malic acid, mangiferin acid, lactic acid, and citric acid were analyzed by HPLC using aqueous phosphoric acid at pH = 2.3 as the aqueous phase and methanol as the organic phase. The experimental conditions were as follows: the column temperature of the C18-XT (4.6 mm × 250 mm × 5 mL) column was 25 °C, and the flow rate was set to be 0.3 mL/min, and the injection volume was 10 µL; ascorbic acid was analyzed by the HPLC using aqueous phosphoric acid at pH = 2.3 as the aqueous phase, and methanol as the organic phase. For ascorbic acid, 0.2% aqueous phosphoric acid was used as the aqueous phase, and methanol was used as the organic phase. The test conditions were as follows: the column temperature of the C18-XT (4.6 mm × 250 mm × 5 mL) column was 25 °C, the flow rate was set at 0.5 mL/min, and the injection volume was 10 µL. The standard curves for the seven measured organic acids are shown in Table 2 below.

#### 2.3.3. HS-GC-IMS Analytical Methods

Headspace gas chromatography–ion mobility spectrometry (HS-GC-IMS) was used for the determination of volatile aroma substances in the soft date kiwifruit resource fruits, and the instrument used in the experiment was a FlavourSpec^®^ Flavour Analyzer. Briefly, 3 g of fruit homogenate was placed in a 20 mL headspace vial, 10 μL of 4-methyl-2-pentanol at 20 ppm was added, and the sample was injected after incubation at 60 °C for 15 min, and three parallel replicates were made for each resource. The chromatographic conditions were as follows (Table 3): the chromatographic column was a WAX column (15 m × 0.53 mm, 1 μm), the column temperature was 60 °C, the carrier gas was N2, and the IMS temperature was 45 °C. The automatic headspace injection conditions were as follows: injection volume was 300 μL, the incubation time was 10 min, the injection needle temperature was 65 °C, the incubation speed was 500 rpm, and the analysis was carried out using 4-methyl-2-pentanol as the internal standard with the concentration of 198 ppb, the signal peak volume of 470.02, and the signal intensity of each signal was about 0.421 ppb. The quantitative calculations were performed according to the following equations.
$$\mathrm{Ci}=\frac{\mathrm{Cis}*\mathrm{Ai}}{\mathrm{Ais}}$$
where Ci is the mass concentration of any component used in the calculation, Cis is the mass concentration of the internal standard used, and Ai/Ais is the volume ratio between any signal peak and the signal peak of the internal standard.

### 2.4. Odor Activity Value (OAV) Calculation

The odor activity value (OAV) was used to evaluate the overall aroma contribution of *A. arguta* fruits. The OAV value was calculated by dividing the concentration of volatile aroma compounds by the odor threshold. The odor thresholds are determined by reference to the *Compilations of Odour Threshold Values in Air, Water and Other Media* (Edition 2011). Volatile aroma compounds with OAV > 1 were considered to be aromatically active and contribute significantly to the overall aroma of the samples.

### 2.5. Data Processing

Excel 2016 was used to organize the experimental data statistically, analysis of variance (ANOVA) was performed by SPSS (version 23.0, IBM, Armonk, NY, USA), and statistical analyses of variance were performed on the experimental data to check for significant differences in the individual results, and all the data were expressed as mean±standard deviation, with *p* < 0.05 indicating significant differences.

The HS-GC-IMS results were analyzed using the Volatile Organic Compounds Analysis Software (VOCal) accompanying the FlavourSpec^®^ Flavour Analyzer, and the volatile aroma compounds were qualitatively analyzed using the retention index database of NIST and the migration time database of IMS built into the GC×IMS Library Search software; the GC-IMS detection was performed by using Savitzky–Golay to perform the smoothing and denoising process, and the migration time normalization method was used by locating the RIP position at position 1, which means that the actual migration time was divided by the peak time of the RIP. The Reporter plug-in was used to compare spectral differences between samples directly, and the Gallery Plot plug-in was used for fingerprinting to visually compare differences in volatile aroma compounds between fruits from different soft date kiwifruit sources. OPLS-DA and VIP values were analyzed using Simca software, and PCA, heatmap, and correlation analyses were performed using the OmicShare tool (https://www.omicshare.com/tools/, accessed on 19 September 2023).

## 3. Results and Analysis

### 3.1. Analysis of Soluble Sugar Content, Titratable Acid Content, and Sugar–Acid Ratio of Fruits from Different A. arguta Resources

Analysis of the differential results (Table 4) showed differences in soluble sugar content, titratable acid content and sugar–acid ratio between fruits of different *A. arguta* resources. The variation of soluble sugar content was 2.94–13.97%, the resource with the highest content was S26, which was significantly higher than the other resources, and the lowest resource was S4; the highest titratable acid content was S24 and S26 with 1.59% and 1.51%, respectively, and the lowest content was S35 with 0.32%. Fruit flavor is largely influenced by the levels of sugars and acids in the fruit. A good flavor requires a high sugar content and a suitable sugar–acid ratio. If the acidity is too high, the fruit may not be palatable. If the sugar content is high but the acidity is too low, the flavor may be bland and lack the balance of sweetness and sourness. If both the sugar and acid levels are too low, the fruit may taste watery and insipid [40]. The sugar–acid ratio of 35 *A. arguta* resource fruits was 2.45–28.50, with S35 having the highest sugar–acid ratio but the lowest titratable acid content, resulting in a more homogeneous flavor. In contrast, S12 has a higher sugar–acid ratio with titratable acid content at higher sugar content; therefore, its fruit flavor can be superior to its source.

### 3.2. Analysis of Organic Acid Content in Fruits of Different A. arguta Resources

The type and content of organic acids affect the acidity of *A. arguta* fruits and the texture of *A. arguta* products, and the content of organic acids varies among different resources (Table 5). Organic acid is an essential component of the fruit and an essential factor affecting fruit quality [41]. The highest oxalic acid content was 0.182 g/L for S12, and the lowest was 0.013 g/L for S31. Oxalic acid, as a ubiquitous component in plants, has long been recognized as a metabolic end product with no obvious physiological role, but from the perspective of food nutrition and human health, long-term consumption of oxalic-acid-rich fruits and vegetables not only reduces the effectiveness of calcium and trace elements in the body but also causes the human body to suffer from renal calculi, diseases of the oral and digestive tracts, and so on [42]. The malic acid in fruits inhibits bacterial damage to the pulp and facilitates fruit preservation [43,44], and S8 had the highest malic acid content of 2.868 g/L, while the lowest content of S5 was 0.212 g/L. Quinic acid and shikimic acid will directly affect the bitter taste of the fruit and are intermediate products of the aromatic substance synthesis pathway, thus indirectly affecting the quality of the fruit [45]; the highest content of quinic acid was S2, 11.426 g/L, which was significantly higher than the other resources, and the lowest content was S17, 1.64 g/L. The shikimic acid content was 0.018–0.093 g/L. Lactic acid was detected in the fruits of some *A. arguta* resources, with the highest level of 0.329 g/L in S1 and the lowest level of 0.015 g/L in S9. Citric acid is characterized by producing acidity quickly and for a sustained period time, and it is capable of causing changes in the threshold of taste substances such as sweetness, sourness, astringency, and bitterness [46]. The citric acid content in the fruits of 35 *A. arguta* resources was 1.987–10.823 g/L, and the resource with the highest content was S2, which was significantly higher than the other resources. Ascorbic acid is widely present in plant tissues and has strong antioxidant properties and a variety of biological functions, such as resistance to stress and disease, but it also can be used for post-harvest storage for horticultural tea growers [47]. The resource with the highest content of ascorbic acid, S5, was 904.739 g/L, which was significantly higher than the other resources, and the content of S7 was the lowest, which was 28.740 g/L. The 35 *A. arguta* resource fruits could be categorized into citric-acid-dominant and quinic-acid-dominant types.

The results of hierarchical clustering analysis (HCA) can better respond to the characteristics of organic acid substances in the fruit samples of different *A. arguta* resources; according to the organic acid cluster analysis of each resource, it can be seen that when the value of the transverse tangent line is taken between 200 and 400 (Figure 1), the 35 *A. arguta* resource fruit samples are divided into six classes: the first class is S5 and S25; the second class is S4, S10, and S21; the third category is S27, S9, and S20; the fourth category is 9 resources, such as S35 and S3; the fifth category is S8, S28, S31, S18, and S214; and the sixth category is 13 resources, such as S22 and S23, which indicates that the samples contained in each category have similarity in organic acids when the value of the transversal line is taken between 200 and 400, and the results of which also show better clustering of fruit samples from different resources of *A. arguta* resources.

### 3.3. HS-GC-IMS Analysis of Fruits from Different A. arguta Resources

The aroma description of *A. arguta* fruits is one of the critical determinants of their quality, and their flavor is also an essential factor in determining whether they are acceptable to consumers [48]. The type and content of volatile compounds and their interactions are the main factors affecting the quality of *A. arguta* fruits. Gas chromatography–mass spectrometry (HS-GC-IMS) is commonly used to characterize and quantify volatile compounds in food [49].

#### 3.3.1. Two-Dimensional Mapping of Volatile Aroma Substances in Fruits of Different *A. Arguta* Resources

There were differences in the two-dimensional mapping profiles of volatile aroma substances of 35 *A. arguta* resources (Figure 2). The differences were mainly reflected in the content, and the color represented the concentration of the substances, with white representing a low concentration of the substances, red representing a high concentration of the substances, and darker colors representing a higher concentration of the substances. The volatile aroma substances in the 35 *A. arguta* resource fruits were well separated by HS-GC-IMS, and the differences between individual samples could be seen.

#### 3.3.2. Comparative Pattern Spectrum of Differences in Volatile Aroma Substances of Fruits from Different *A. arguta* Resources

HS-GC-IMS was used to obtain full information on the volatiles in the fruits of the *A. arguta* resource, and difference comparison mode spectra represented the differences between the samples. The horizontal and vertical axes of the difference plots represent the ionic migration time of the volatile compounds and the retention time at the ionic peaks of the reactants, respectively, and each point represents the monomer of the volatile compounds extracted from the samples or their dimers [50]. Taking S1 as a reference (Figure 3), the rest of the spectrum subtracts the signal peaks in S1 to obtain the difference comparison mode spectrum between the two. The red area in the graph indicates that the concentration of the substance in this sample is higher than that of S1, and the blue area indicates that the attention of the substance in this sample is lower than that of S1. The white area indicates that the attention of the substance in this sample is comparable to that of S1. Differential mapping analysis showed that S1 contained higher levels of hexyl propanoate, ethyl (E)-hex-2-enoate, ethanol, isobutanol, hexanal, and trans-2-hexenal than some of the resource fruits.

#### 3.3.3. Identification of Substances

For the qualitative analysis of various volatiles in the *A. arguta* resource fruit samples, the drift times and RIs in the IMS were compared to authentic controls. Subsequently, we identified 97 signal peaks (including monomers and dimers) from the two-dimensional profiles, and 76 volatile aroma substances were initially identified, as shown in Table 6. These contain 18 esters, 14 alcohols, 16 ketones, 12 aldehydes, seven terpenoids, three pyrazines, two furans, two acids and two other compounds, which essentially cover the range of aroma compounds found in fruits [19,51,52,53]. Nineteen of these substances, including methyl butanoate, isoamyl acetate, ethyl hexanoate, ethyl acetate, carveol, 1-hexanol, cineole, and 2-heptanone, formed dimers, which was related to the concentration of the volatile aroma substances and their proton affinity. The transfer of protons from reactants with higher proton affinity than water in highly concentrated substances to substances with higher proton affinity thus contributes to the formation of dimers [54].

#### 3.3.4. Fingerprint Analysis of Volatile Components of Fruits from Different *A. arguta* Resources

Although difference mapping shows overall differences in flammable substances in fruits from different *A. arguta* sources, fingerprinting can more accurately identify differences in the nature and concentration of individual substances. In fingerprint mapping, each row represents the overall signal peak of a sample, and each column represents the same substance in a different model. Color refers to the content of volatile substances; the brighter the color, the higher its content. As shown in Figure 4, the volatile compounds with high variability among the *A. arguta* resource fruit samples were methyl acetate, hexyl propanoate, hexyl acetate, ethyl hexanoate-D, ethyl isovalerate, butyl acetate-D, citronellyl formate, cineole-D, 2-heptanol, 2-octanone, 2-butanone, 3,5-dimethyl-1,2-cyclopentanedione, butanal, isovaleraldehyde, (Z)-4-heptenal, myrcene, and 2-methoxy-3-methylpyrazine.

### 3.4. Analysis of the Relative Content of Volatile Components

#### 3.4.1. Esters

Ester compounds are the most diverse compounds detected in each resource (Figure 5), which mainly reflect fruity and floral aromas [55]; among the ester compounds detected, methyl butanoate, ethyl acetate, butyl acetate, and ethyl hexanoate have apple and pineapple aromas, ethyl butyrate has a floral aroma, and isoamyl acetate has a sweet aroma. The relative content of esters in 35 resource fruits was 2142.40–6065.74 ppb, accounting for 12.91–30.22% of the total volatiles, of which the relative content of esters in S34 was the highest. The content of ethyl propanoate was the highest among the ester compounds detected in the 35 resource fruits. It best reflected the fruity flavor of *A. arguta* fruits.

#### 3.4.2. Alcohols

The percentage of alcohols was 8.78–21.45% (Figure 5), and their aroma was mainly grassy and alcoholic. The highest relative content of alcohols was S18, with 4420.72 ppb, followed by S24, with 3126.88 ppb, and the lowest relative content was S33, with 1520.96 ppb. Thirty-five *A. arguta* resource fruits were detected with a higher content of isobutanol and 1-hexanol among the alcohols, which best reflected the grassy aroma of *A. arguta* fruits.

#### 3.4.3. Ketones

The content of ketones was 1581.99–6614.19 ppb, accounting for 8.50–32.95% of the total volatile compounds (Figure 5). The resource with the highest content was S34, and the lowest was S27. The ketones detected in the fruits of 35 *A. arguta* resources were more elevated in 2-heptanone and hydroxyacetone, with 2-heptanone having a banana aroma and slight medicinal flavor.

#### 3.4.4. Aldehydes

Aldehydes were the compounds with the highest relative content detected in the 34 samples except S34, which was similar to the results of Sun Yang [14] et al. at 3480.11–11746.16 ppb, with the highest resource being S6 and the lowest S34, and the content of aldehydes in each sample accounted for 17.34–58.38% of the total volatiles (Figure 5). The highest relative content of aldehydes detected in the fruits of the resources was trans-2-hexenal, which was mainly characterized by grassy, apple, and aldehydic aromas.

#### 3.4.5. Other Compounds

Compounds such as terpenoids, acids, pyrazines, and furans were also detected in the fruits of the *A. arguta* resource, all in low relative amounts, accounting for 2.22–8.51%, 0.65–2.27%, 0.26–1.86%, and 0.64–2.00% of the total volatile compounds, respectively (Figure 5).

### 3.5. Principal Component Analysis of Fruit Samples from Different A. arguta Resources

In order to better present and differentiate between fruit samples from different *A. arguta* resources, volatile compounds identified by HS-GC-IMS were analyzed by PCA. Unsupervised multidimensional statistics (PCA) were used to determine the samples to distinguish the magnitude of variation among different sample groups, subgroups, and within-group samples of fruits from various *A. arguta* resources. The contribution rate of PC1 was 29.2%, and that of PC2 was 13.1%, with the 35 groups of samples showing a clear tendency to segregate on the two-dimensional plots, and the magnitude of variation of the samples within the groups was obvious. The principal component results (Figure 6) showed significant overall differences in the aroma substances of the 35 groups of samples and differentiated them. As shown in Figure 6, the magnitude of intra-group variation was more significant for S14, S23, and S15, and the distance of the aroma characteristics of S14, S34, S35, S31, S32, S2, and S33 was farther away from each other, indicating that there were significant differences in the aroma characteristics among the different samples.

### 3.6. OPLS-DA Analysis and the Model Validation of Volatile Aroma Compounds of A. arguta Resource Fruits

OPLS-DA is a supervised discriminant statistical method that not only realizes the identification of sample differences but also obtains the characteristic markers of sample differences [56]. The contribution of each variable to the aroma of *A. arguta* was further quantified based on the variable importance (VIP) in the OPLA-DA model, and the volatile aroma compounds with VIP values greater than 1 were screened as the main characteristic volatile markers [57]. With 76 volatile aroma substances as dependent variables and different *A. arguta* resources as independent variables, effective differentiation of *A. arguta* fruit samples from 35 resources could be achieved by OPLS-DA (Figure 7A). The fit index (RX2) for the independent variable in this analysis was 0.987, the fit index (RY2) for the dependent variable was 0.793, and the model prediction index (Q2) was 0.554, with R2 and Q2 exceeding 0.5 to indicate acceptable model fit results [58]. After 200 replacement tests, as shown in Figure 7B, the intersection of the Q2 regression line with the vertical axis was less than 0, indicating that there was no overfitting of the model and validating the model, and it was considered that the results could be used for the identification and analysis of volatile aroma compounds in the fruits of different *A. arguta* resources.

The aroma quality of soft date kiwifruit fruit depends on the result of the joint action of several volatile aroma compounds; according to the criteria of *p* < 0.05 and VIP > 1, 33 kinds of *A. arguta* resource fruit volatile aroma substances were screened out as the main aroma substances (Figure 8), among which there are eight kinds of esters, five kinds of alcohols, six kinds of ketones, six kinds of aldehydes, two kinds of acids, three kinds of terpenoids, one kind of furan, and two kinds of other compounds.

### 3.7. OAV Analysis of the Main Aroma Components of Fruit Samples from Different A. arguta Resources

Although HS-GC-IMS characterized and quantified the volatile aroma substances of *A. arguta* resource fruits and OPLS-DA can screen potential characteristic volatile markers of volatile aroma substances of *A. arguta* resource fruits, the level of volatile aroma substance content does not determine the aroma contribution of each substance. Consumers usually judge the acceptability of food by aroma and flavor [59]. The odor activity of volatile compounds in *A. arguta* fruits is one of the main sensory characteristics that determine the quality of the fruit. OAV can reflect the contribution of individual volatile aroma compounds to the characteristic flavor of the sample. The OAV of volatile aroma compounds depends on their concentration and odor threshold. Based on previous studies, it was shown that volatile aroma compounds with OAV>1 contributed more to the overall aroma of the samples, and the larger the OAV value, the greater the contribution of the compound [60]. In this study, the volatile aroma compounds screened by OPLS-DA with VIP values greater than 1 were analyzed for OAV, and a total of 18 volatile aroma compounds with OAV > 1 were detected according to the calculation (Appendix A), among which six types of esters were esters, namely methyl butanoate, isoamyl acetate, hexyl propanoate, butyl acrylate, butyl isovalerate and 1-methoxy-2-propyl acetate; three types of alcohols, namely 3-methyl-1-butanol, 1-hexanol, and leaf alcohol; three types of ketones, namely l(-)-Carvone, 5-methyl-3-heptanone, and 3,4-dimethyl-1,2-cyclopentanedione; three types of aldehydes, namely heptanal, butanal, and isovaleraldehyde; and three types of terpenes, namely dipentene, alpha-pinene, and terpinolene. Although the OAV values of the 35 *A. arguta* samples varied, in comparison, isoamyl acetate, 3-methyl-1-butanol, 1-hexanol, and butanal had higher OAV values than the other compounds, ranging from 183.09 to 1175.54, 10.19 to 6.98, 33.55 to 126.40, and 30.42 to 90.93, respectively, suggesting that the contribution of these four volatile compounds to the overall kiwifruit aroma was greater. Isoamyl acetate had a fruity, sweet, and floral aroma; 3-methyl-1-butanol had an alcoholic and fruity aroma; and 1-hexanol had a grassy, fruity, sweet, and alcoholic aroma, which are essential aromatic characteristics in *A. arguta* fruits.

### 3.8. Heat Map Analysis, PCA Analysis and Correlation Analysis of Volatile Aroma Compounds with OAV > 1 in Fruits of Different A. arguta Resources

Concentrations of aroma substances with OAV greater than 1 in volatile compounds from 35 *A. arguta* resource fruit samples were clustered using hierarchical analysis, and similarity was calculated using Pearson. Based on the heat map analysis of the samples (Figure 9), the red color indicates the high expression of the volatile aroma compound in the embodiment, and the blue color indicates the low expression of the volatile aroma compound in the selection, which can clearly show the differences between the concentrations of each substance in different *A. arguta* resources.

Volatile aroma compounds with OAV values greater than 1 were analyzed in the PCA of the fruits of *A. arguta* resources (Figure 10). The contribution of PC1 was 20.7% and the contribution of PC2 was 13.6%. The PCA scatters of most of the samples were dispersed, indicating that the similarity between these samples was low. Few samples are distributed in the second quadrant, only S5, S31, and S32, and the distribution is more dispersed. The scatters of the samples distributed in the center of the axes are more clustered, indicating higher similarity between them.

A significant correlation between substances is indicated by a correlation coefficient between 0.8 and 1.0, a strong correlation is indicated by a correlation coefficient between 0.6 and 0.8, a moderate correlation is indicated by a correlation coefficient between 0.4 and 0.6, a weak correlation is indicated by a correlation coefficient between 0.2 and 0.4, and correlation coefficients between 0 and 0.2 indicate that there is no correlation between the substances or that the correlation is very weak. As can be seen in Figure 11, there is a highly significant correlation between 1-hexanol and leaf alcohol and a strong correlation between 1-methoxy-2-propyl acetate and heptanal, alpha-pinene, and terpinolene. Moderate correlations were found between methyl butanoate and hexyl propanoate, 1-methoxy-2-propyl acetate and isovaleraldehyde, 1-hexanol and isovaleraldehyde, and dipentene and alpha-pinene. A strong negative correlation was found between methyl butyrate and heptanal.

## 4. Conclusions

*Actinidia arguta*, a type of kiwifruit, has good organoleptic quality and rich nutritional value. Therefore, it is important to study its flavor quality and volatile aroma components. This study used 35 A. arguta resource fruits as materials to measure and analyze their soluble sugar, titratable acid, and sugar–acid ratio. The results showed that the soluble sugar content of 35 *A. arguta* resource fruits was 2.94–13.97%, the content of titratable acid was 0.32–1.59%, and the sugar–acid ratio was 2.45–28.50. In contrast, S12 had a higher sugar–acid ratio with a higher titratable acid content and a higher sugar content, which indicated a superior fruit flavor compared to its source. High-performance liquid chromatography (HPLC) was used to determine the content of organic acids. The results showed that the 35 fruits could be classified into two types: citric-acid-dominant and quinic-acid-dominant. Lactic acid was also detected in some of the fruits.

Headspace gas chromatography–ion mobility spectrometry (HS-GC-IMS) was used to analyze the volatile aroma substances of different *A. arguta* resources, and a total of 76 volatile aroma substances were identified, which contained 18 esters, 14 alcohols, 16 ketones, 12 aldehydes, seven terpenes, three pyrazines, two furans, two acids, and two other compounds, and these compounds basically covered the types of aroma compounds in the fruit. With 76 volatile aroma substances as the dependent variables and different soft date kiwifruit resources as the independent variables, 33 volatile aroma substances with VIP > 1 were screened out as the main aroma substances of *A. arguta* resource fruits by OPLS-DA analysis. The volatile aroma compounds screened by OPLS-DA with VIP values greater than 1 were subjected to OAV analysis, and 18 volatile aroma compounds with OAV>1 were screened based on the calculation of their odor activity values, including six esters, three alcohols, three ketones, three aldehydes, and three terpenoids. Comparison of the OAV values revealed that isoamyl acetate, 3-methyl-1-butanol, 1-hexanol, and butanal had higher OAV values than the other compounds, indicating that these four volatile compounds were the main contributors to the overall aroma of *A. arguta*. Headspace gas chromatography–ion mobility spectrometry can show the commonalities and differences between the samples, which makes up for the perceived inadequacy of sensory evaluation and plays a useful and complementary role in the evaluation of the flavor quality of *A. arguta*. This provides a theoretical basis for screening *A. arguta* resources with excellent flavor quality, enhancing and improving the flavor quality of *A. arguta* processed products, and at the same time, provides a theoretical basis for the scientific understanding of the characteristic compounds of fruit aroma of different *A. arguta*. However, the IMS database is not complete enough, which prevents some compounds isolated by GC from being characterized. Therefore, the gradual enrichment of the IMS database is an important development direction for the detection of volatile aroma compounds in the future. At the same time, it is necessary to further combine the nutritional quality and volatile flavor quality to establish a more detailed evaluation system of *A. arguta* quality to lay a theoretical foundation for the development of excellent *A. arguta* resources.

## Figures and Tables

**Figure 1 foods-12-03615-f001:**
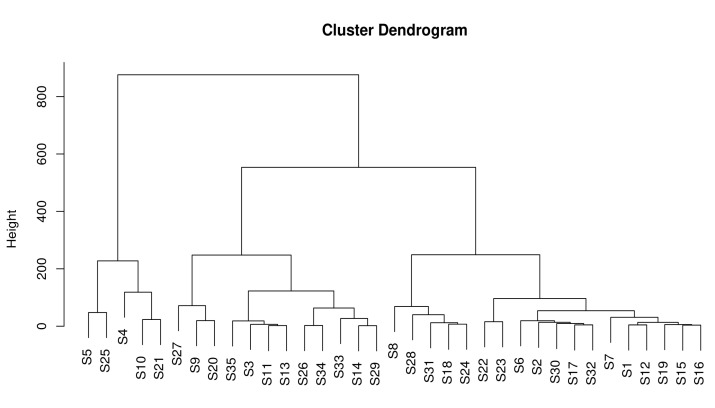
Hierarchical cluster analysis of organic acid content in fruits of different *A. arguta* resources.

**Figure 2 foods-12-03615-f002:**
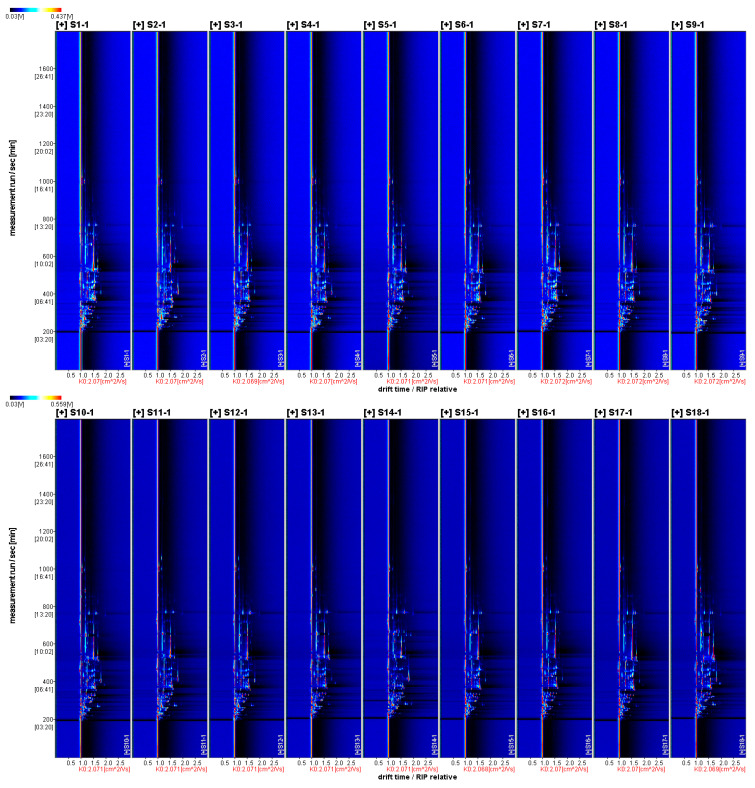
HS-GC-IMS 2D mapping (top view).

**Figure 3 foods-12-03615-f003:**
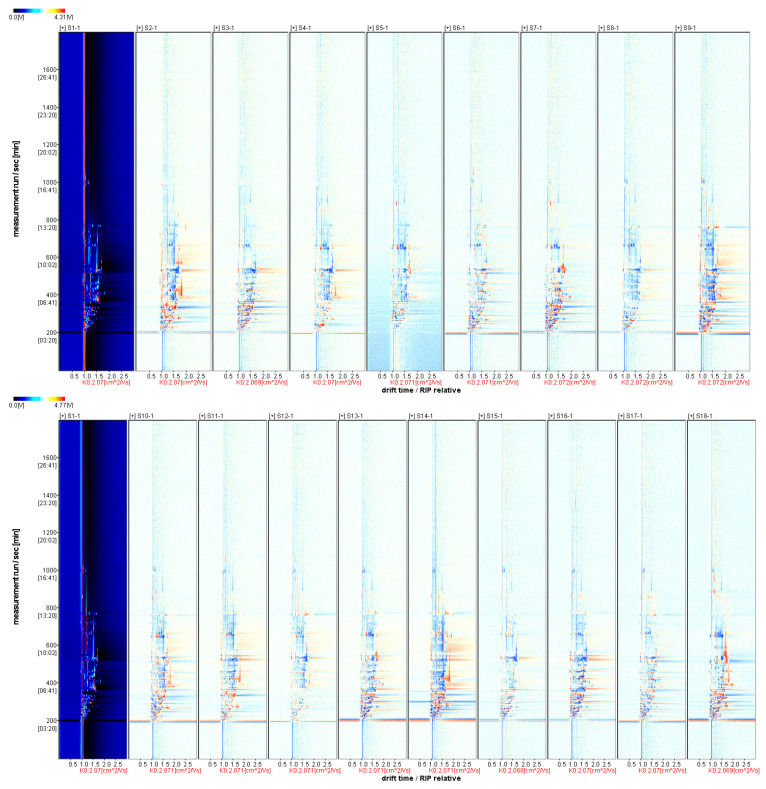
HS-GC-IMS difference comparison mode spectra.

**Figure 4 foods-12-03615-f004:**
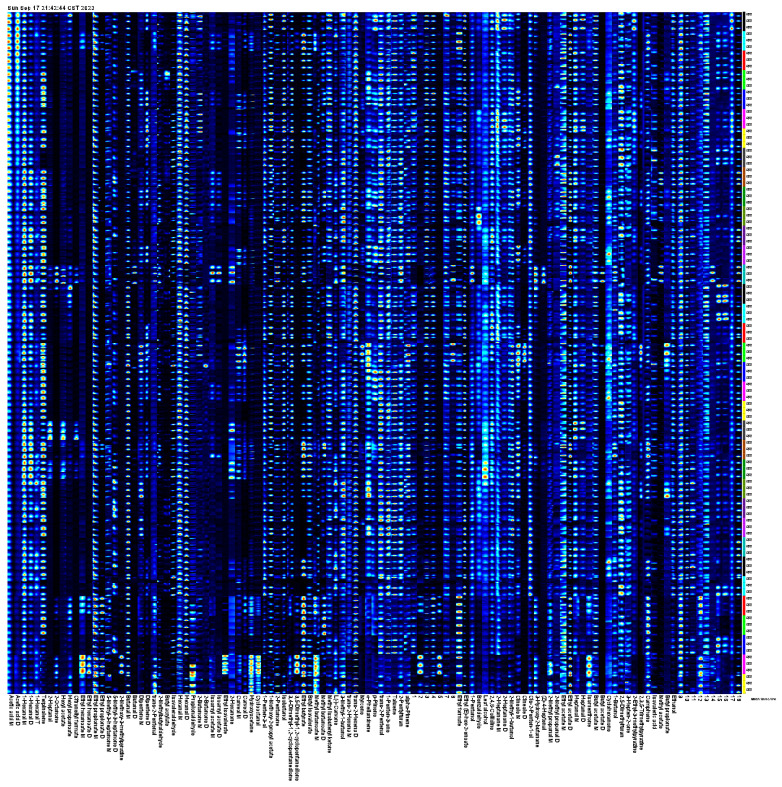
Fingerprints of volatile compounds in fruits of different *A. arguta* resources.

**Figure 5 foods-12-03615-f005:**
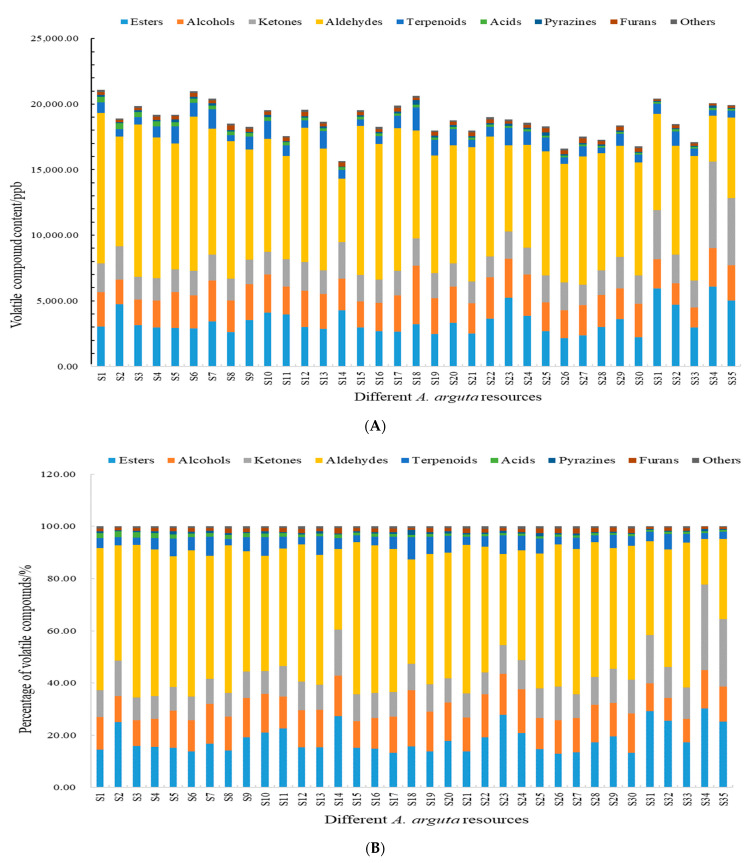
Content (**A**) and percentage (**B**) of volatile compounds in different *A. arguta* resources.

**Figure 6 foods-12-03615-f006:**
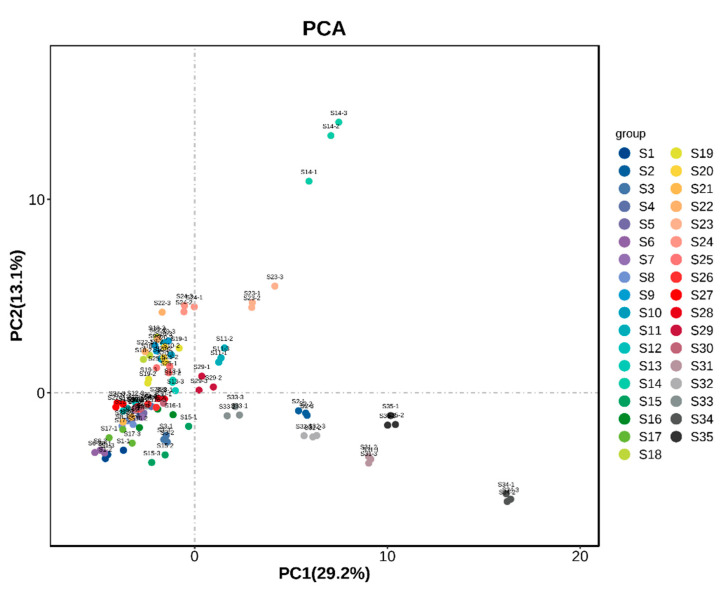
Principal component analysis of fruit samples from different *A. arguta* resources.

**Figure 7 foods-12-03615-f007:**
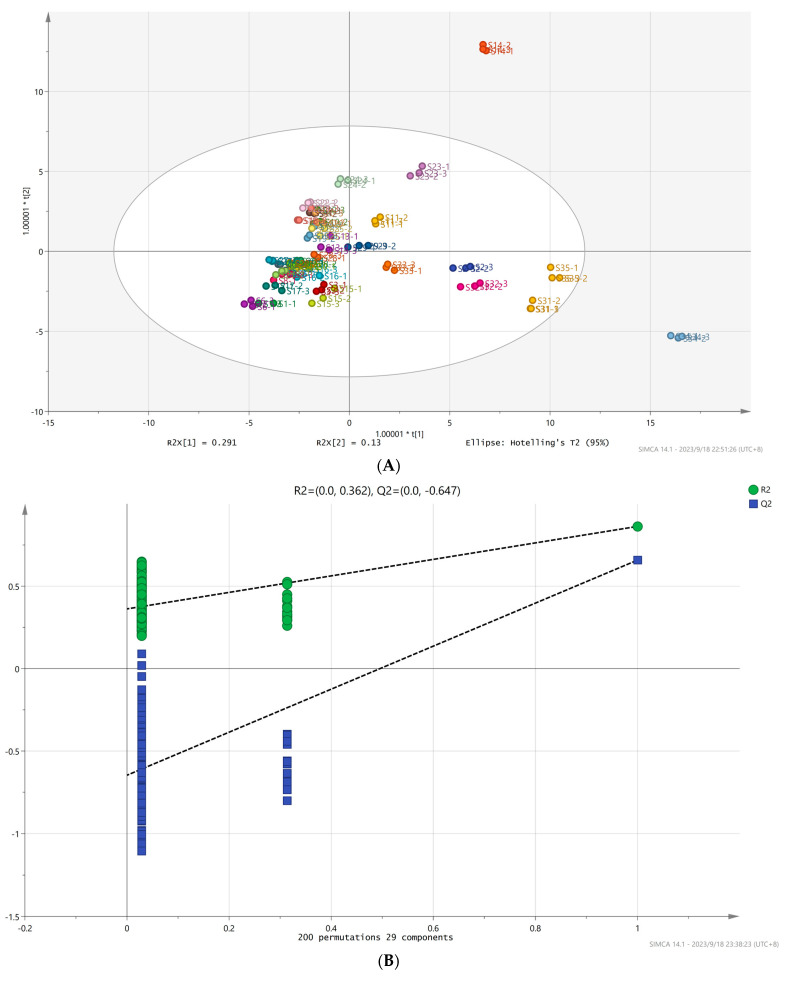
OPLS-DA of volatile aroma compounds in fruits of different *A. arguta* resources (**A**) and model cross-validation results (**B**).

**Figure 8 foods-12-03615-f008:**
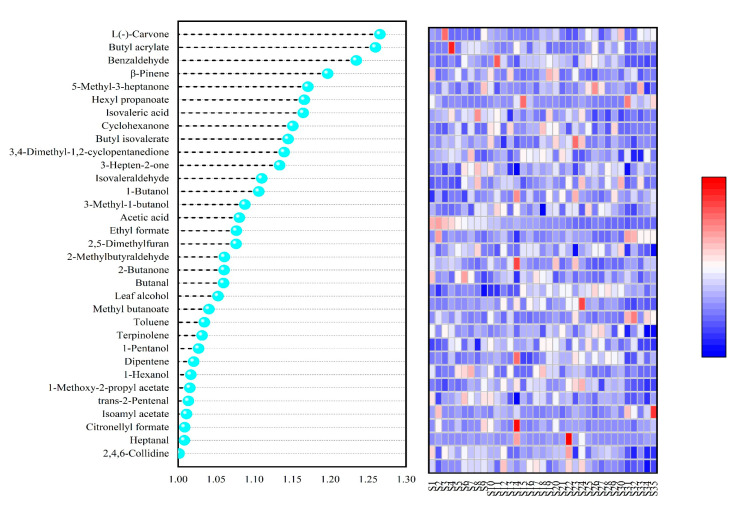
OPLS-DA analysis of VIP values of major volatile aroma substances in fruits of different *A. arguta* resources.

**Figure 9 foods-12-03615-f009:**
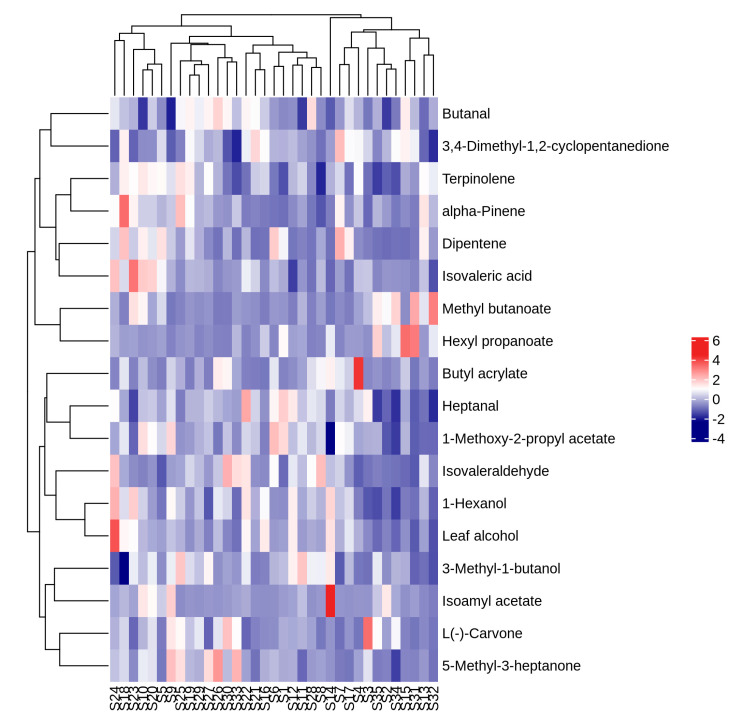
Clustering heat map analysis of volatile aroma compounds with OAV greater than 1 in fruits of different *A. arguta* resources.

**Figure 10 foods-12-03615-f010:**
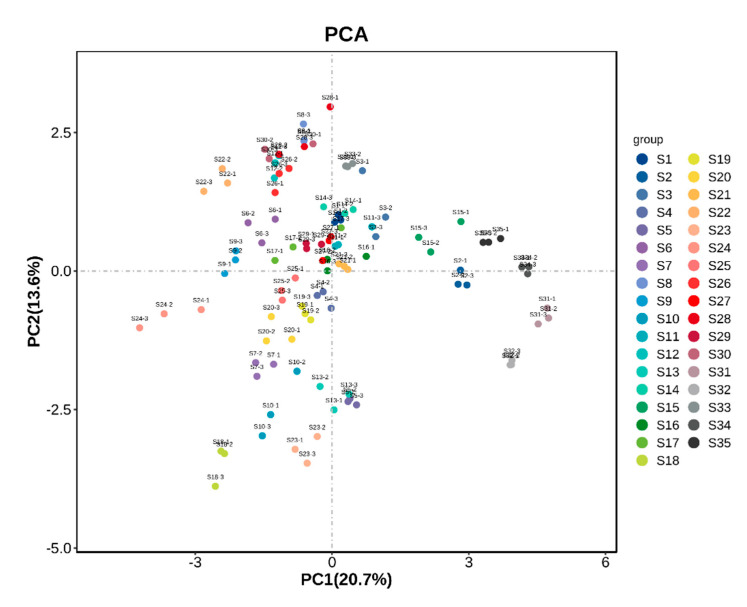
Scatter plot of PCA analysis of volatile aroma compounds with OAV greater than 1 in fruits of different *A. arguta* resources.

**Figure 11 foods-12-03615-f011:**
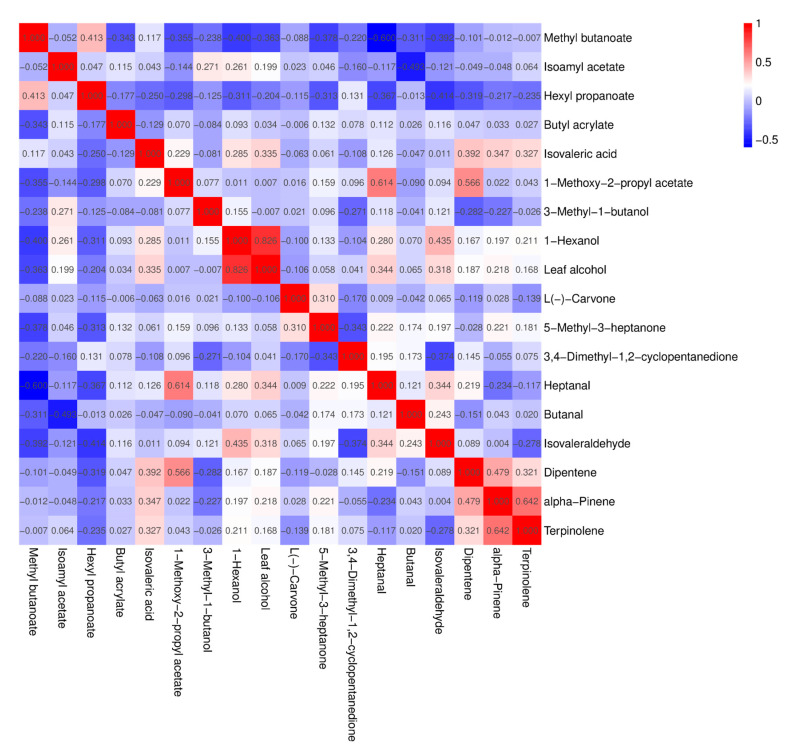
Correlation analysis of volatile aroma compounds with OAV greater than 1 in fruits of different *A. arguta* resources.

**Table 1 foods-12-03615-t001:** Resources and sources of 35 *A. arguta*.

No.	Name	Source	No.	Name	Source	No.	Name	Source
S1	A020203	Fusong County, Jilin Province, China	S13	A130701	Ji’an County, Jilin Province, China	S25	B080401	Ji’an County, Jilin Province, China
S2	A040103	Ji’an County, Jilin Province, China	S14	A130801	Ji’an County, Jilin Province, China	S26	B080701	Ji’an County, Jilin Province, China
S3	A060902	Zuojia Town, Jilin Province, China	S15	A140101	Zuojia Town, Jilin Province, China	S27	T040501	Fusong County, Jilin Province, China
S4	A100101	Ji’an County, Jilin Province, China	S16	A140301	Zuojia Town, Jilin Province, China	S28	T060203	Ji’an County, Jilin Province, China
S5	A100703	Ji’an County, Jilin Province, China	S17	A140602	Dunhua City, Jilin Province, China	S29	T060301	Fusong County, Jilin Province, China
S6	A100801	Ji’an County, Jilin Province, China	S18	A160701	Zuojia Town, Jilin Province, China	S30	T060503	Ji’an County, Jilin Province, China
S7	A101201	Dunhua City, Jilin Province, China	S19	A170303	Fusong County, Jilin Province, China	S31	SH1	Zuojia Town, Jilin Province, China
S8	A111001	Zuojia Town, Jilin Province, China	S20	A180303	Fusong County, Jilin Province, China	S32	SH2	Zuojia Town, Jilin Province, China
S9	A120403	Dunhua City, Jilin Province, China	S21	A180902	Zuojia Town, Jilin Province, China	S33	SH3	Zuojia Town, Jilin Province, China
S10	A120601	Dunhua City, Jilin Province, China	S22	A191002	Ji’an County, Jilin Province, China	S34	SH4	Zuojia Town, Jilin Province, China
S11	A130101	Ji’an County, Jilin Province, China	S23	B020802	Zuojia Town, Jilin Province, China	S35	SH5	Zuojia Town, Jilin Province, China
S12	A130602	Ji’an County, Jilin Province, China	S24	B070101	Zuojia Town, Jilin Province, China			

**Table 2 foods-12-03615-t002:** The organic acid standard curve.

Name	Concentration g/L	Standard Curves	R^2^
Oxalic acid	1.02	y = 24763x − 735.65	0.9998
Quinic acid	1.01	y = 779.46x − 18.648	0.9962
Malic acid	1.00	y = 1613.5x − 7.0785	0.9999
Shikimic acid	1.00	y = 45865x + 2285.4	0.9977
Lactic acid	1.08	y = 1272.5x − 4.6931	1
Citric acid	1.02	y = 2028.3x − 18.753	0.9999
Ascorbic acid	1.03	y = 24.297x − 41.339	0.9998

**Table 3 foods-12-03615-t003:** Gas chromatography conditions.

Time (min: sec)	E1 (Drift Gas)	E2 (Carrier Gas)	Recording
00:00,000	150 mL/min	2 mL/min	rec
02:00,000	150 mL/min	2 mL/min	-
10:00,000	150 mL/min	10 mL/min	-
20:00,000	150 mL/min	100 mL/min	-
30:00,000	150 mL/min	100 mL/min	stop

**Table 4 foods-12-03615-t004:** Content of soluble sugar, titratable acid and sugar–acid ratio of different *A. arguta* resources.

Name	Soluble Sugar %	Titratable Acid %	Sugar–Acid Ratio
S1	5.74 ± 0.34 opq	0.78 ± 0.01 k	7.39 ± 0.39 hij
S2	5.35 ± 0.08 q	1.00 ± 0.06 ij	5.34 ± 0.23 op
S3	6.66 ± 0.17 jkl	0.98 ± 0.08 j	6.80 ± 0.47 ijkl
S4	2.94 ± 0.12 u	1.20 ± 0.04 ef	2.45 ± 0.17 t
S5	5.84 ± 0.24 nop	0.82 ± 0.02 k	7.10 ± 0.38 hijk
S6	5.34 ± 0.09 q	0.97 ± 0.07 j	5.51 ± 0.48 nop
S7	6.49 ± 0.42 klm	0.85 ± 0.04 k	7.63 ± 0.21 hi
S8	7.04 ± 0.16 ij	1.39 ± 0.09 c	5.08 ± 0.30 pq
S9	4.14 ± 0.03 s	0.98 ± 0.01 j	4.24 ± 0.04 r
S10	6.65 ± 0.39 jkl	1.00 ± 0.18 ij	6.65 ± 0.87 jkl
S11	7.14 ± 0.33 hi	1.04 ± 0.03 hij	6.89 ± 0.47 ijkl
S12	9.40 ± 0.41 b	0.77 ± 0.05 k	12.21 ± 0.64 e
S13	4.44 ± 0.09 rs	0.64 ± 0.01 l	6.93 ± 0.20 ijkl
S14	6.03 ± 0.23 no	1.05 ± 0.02 hij	5.72 ± 0.15 mnop
S15	7.20 ± 0.48 ghi	1.11 ± 0.08 fgh	6.47 ± 0.86 klm
S16	6.05 ± 0.21 mno	0.95 ± 0.08 j	6.35 ± 0.71 klm
S17	4.85 ± 0.33 r	0.84 ± 0.04 k	5.73 ± 0.62 mnop
S18	8.25 ± 0.10 cd	1.17 ± 0.02 efg	7.05 ± 0.06 ijk
S19	3.48 ± 0.10 t	1.02 ± 0.03 hij	3.43 ± 0.20 s
S20	6.84 ± 0.32 ijk	1.02 ± 0.01 hij	6.71 ± 0.39 jkl
S21	6.82 ± 0.27 ijk	1.26 ± 0.02 de	5.42 ± 0.23 op
S22	8.31 ± 0.30 cd	1.05 ± 0.03 hij	7.92 ± 0.34 h
S23	5.71 ± 0.09 opq	1.09 ± 0.04 ghi	5.24 ± 0.17 o
S24	8.70 ± 0.40 c	1.59 ± 0.06 a	5.48 ± 0.15 op
S25	9.36 ± 0.54 b	1.47 ± 0.01 b	6.35 ± 0.34 klmn
S26	13.97 ± 0.10 a	1.51 ± 0.05 ab	9.0.26 g
S27	5.43 ± 0.29 pq	1.23 ± 0.02 e	4.40 ± 0.24 qr
S28	7.79 ± 0.17 ef	1.04 ± 0.09 hij	7.47 ± 0.60 hij
S29	8.18 ± 0.36 de	1.33 ± 0.04 cd	6.15 ± 0.41 lmno
S30	7.54 ± 0.06 fgh	1.17 ± 0.04 efg	6.46 ± 0.23 klm
S31	8.30 ± 0.06 cd	0.47 ± 0.03 mn	17.78 ± 1.06 b
S32	6.22 ± 0.06 lmn	0.40 ± 0.02 no	15.67 ± 0.39 c
S33	7.62 ± 0.14 fg	0.53 ± 0.02 m	14.47 ± 0.38 d
S34	3.68 ± 0.09 t	0.37 ± 0.01 o	10.04 ± 0.26 f
S35	9.23 ± 0.07 b	0.32 ± 0.01 o	28.50 ± 1.04 a
CV(%)	30.74	32.03	61.25

Means with different letters in the same column express significant differences (Duncan’s test *p* < 0.05).

**Table 5 foods-12-03615-t005:** Content of organic acids in different *A. arguta* resources.

Name	Oxalic Acid g/L	Quinic Acid g/L	Malic Acid g/L	Shikimic Acid g/L	Lactic Acid g/L	Citric Acid g/L	Ascorbic Acid g/L
S1	0.030 ± 0.003 p	7.714 ± 0.318 c	0.765 ± 0.040 st	0.51 ± 0.002 g	0.329 ± 0.0.014 a	8.113 ± 0.051 f	59.617 ± 0.067 x
S2	0.026 ± 0.003 pq	11.426 ± 0.109 a	2.753 ± 0.066 b	0.026 ± 0.002 n	0.208 ± 0.010 b	10.823 ± 0.149 a	67.872 ± 0.063 w
S3	0.133 ± 0.014 efghij	6.085 ± 0.051 g	1.666 ± 0.017 i	0.043 ± 0.004 hi	N.A.	7.890 ± 0.042 g	334.402 ± 15.919 m
S4	0.154 ± 0.014 bcd	5.764 ± 0.039 i	1.591 ± 0.024 j	0.019 ± 0.002 p	0.102 ± 0.003 g	8.642 ± 0.067 d	677.253 ± 0.273 e
S5	0.105 ± 0.011 no	2.872 ± 0.017 v	0.212 ± 0.017 y	0.046 ± 0.002 hi	N.A.	3.479 ± 0.018 x	904.739 ± 0.215 a
S6	0.141 ± 0.013 cedfg	6.682 ± 0.026 e	1.184 ± 0.023 n	0.062 ± 0.002 de	0.046 ± 0.003 m	8.266 ± 0.017 e	82.676 ± 0.195 u
S7	0.156 ± 0.008 bc	5.432 ± 0.018 k	0.684 ± 0.010 u	0.081 ± 0.002 b	0.083 ± 0.002 i	7.267 ± 0.024 j	28.740 ± 0.341 z
S8	0.016 ± 0.001 pq	8.544 ± 0.016 b	2.868 ± 0.014 a	0.073 ± 0.003 c	N.A.	10.547 ± 0.030 b	209.252 ± 0.094 r
S9	0.020 ± 0.001 pq	5.447 ± 0.014 k	1.000 ± 0.011 p	0.043 ± 0.003 hi	0.015 ± 0.002 o	6.735 ± 0.014 l	530.055 ± 0.125 g
S10	0.165 ± 0.010 b	6.378 ± 0.018 f	1.338 ± 0.010 l	0.041 ± 0.004 ij	N.A.	6.793 ± 0.013 k	772.682 ± 0.173 d
S11	0.136 ± 0.017 efghi	4.047 ± 0.014 st	2.174 ± 0.013 e	0.047 ± 0.003 hi	0.060 ± 0.003 kl	5.014 ± 0.019 u	338.561 ± 0.316 m
S12	0.182 ± 0.010 a	5.486 ± 0.013 k	1.924 ± 0.021 f	0.026 ± 0.003 no	0.147 ± 0.008 d	6.837 ± 0.009 k	56.312 ± 0.166 x
S13	0.139 ± 0.013 defgh	5.001 ± 0.013 m	1.697 ± 0.012 hi	0.064 ± 0.002 d	N.A.	3.153 ± 0.012 y	338.813 ± 0.188 m
S14	0.156 ± 0.017 cdef	3.976 ± 0.014 tu	2.428 ± 0.016 c	0.018 ± 0.002 p	N.A.	6.624 ± 0.010 m	393.866 ± 0.133 k
S15	0.020 ± 0.002 pq	6.983 ± 0.009 d	1.744 ± 0.021 g	0.061 ± 0.002 de	0.113 ± 0.004 f	8.076 ± 0.020 f	48.530 ± 0.132 y
S16	0.150 ± 0.015 bcde	4.838 ± 0.010 no	1.230 ± 0.015 m	0.057 ± 0.002 fg	N.A.	5.680 ± 0.024 r	46.768 ± 0.093 y
S17	0.092 ± 0.011 o	1.641 ± 0.013 w	0.302 ± 0.014 x	0.091 ± 0.002 a	0.072 ± 0.002 j	1.987 ± 0.007 z	65.416 ± 0.071 w
S18	0.122 ± 0.008 hijklm	2.871 ± 0.014 v	0.514 ± 0.031 v	0.084 ± 0.003 b	N.A.	3.136 ± 0.021 y	244.467 ± 0.093 p
S19	0.018 ± 0.001 pq	4.615 ± 0.011 q	2.227 ± 0.014 d	0.034 ± 0.003 kl	0.137 ± 0.005 e	8.832 ± 0.022 c	51.441 ± 0.078 y
S20	0.115 ± 0.100 bc	5.901 ± 0.018 h	0.982 ± 0.009 p	0.061 ± 0.005 ef	N.A.	6.486 ± 0.012 n	510.680 ± 0.105 h
S21	0.106 ± 0.007 mno	3.902 ± 0.036 uv	0.920 ± 0.007 q	0.018 ± 0.002 p	0.070 ± 0.003 j	4.362 ± 0.014 w	795.696 ± 0.217 c
S22	0.138 ± 0.027 ijklm	5.828 ± 0.011 hi	1.706 ± 0.013 h	0.035 ± 0.003 kl	N.A.	6.138 ± 0.027 p	125.166 ± 0.182 s
S23	0.016 ± 0.002 pq	7.082 ± 0.005 d	0.903 ± 0.010 q	0.075 ± 0.003 c	N.A.	5.215 ± 0.011 t	109.725 ± 0.074 t
S24	0.142 ± 0.008 cdefg	6.447 ± 0.014 f	0.776 ± 0.010 st	0.054 ± 0.003 fg	0.094 ± 0.003 h	6.291 ± 0.017 o	249.555 ± 0.096 o
S25	0.118 ± 0.010 jklnm	5.647 ± 0.008 j	1.180 ± 0.031 n	0.028 ± 0.002 mn	N.A.	7.695 ± 0.014 i	857.254 ± 0.061 b
S26	0.096 ± 0.010 o	5.312 ± 0.040 l	1.522 ± 0.008 k	0.067 ± 0.003 d	0.054 ± 0.004 l	7.761 ± 0.009 h	457.152 ± 0.089 i
S27	0.173 ± 0.006 pq	4.560 ± 0.007 q	1.500 ± 0.007 k	0.035 ± 0.002 jk	0.061 ± 0.003 k	7.945 ± 0.022 g	582.221 ± 0.123 f
S28	0.146 ± 0.027 fghijk	5.665 ± 0.013 j	0.850 ± 0.012 r	0.049 ± 0.004 h	N.A.	5.176 ± 0.012 t	277.654 ± 0.143 n
S29	0.119 ± 0.007 ijklmn	4.292 ± 0.013 r	1.099 ± 0.027 o	0.046 ± 0.002 hi	0.161 ± 0.002 c	5.595 ± 0.014 s	393.842 ± 0.131 k
S30	0.113 ± 0.007 lmn	4.092 ± 0.009 s	0.744 ± 0.012 t	0.043 ± 0.006 hi	N.A.	3.034 ± 0.013 y	74.301 ± 0.137 v
S31	0.013 ± 0.002 q	4.748 ± 0.024 op	0.295 ± 0.010 x	0.037 ± 0.003 kl	0.027 ± 0.002 n	5.231 ± 0.023 t	237.644 ± 0.058 q
S32	0.115 ± 0.010 klmn	4.602 ± 0.021 q	0.289 ± 0.010 x	0.027 ± 0.002 n	0.033 ± 0.002 n	5.218 ± 0.027 t	66.265 ± 0.064 w
S33	0.128 ± 0.011 ghijkl	4.667 ± 0.040 pq	0.795 ± 0.017 s	0.038 ± 0.003 jk	N.A.	5.619 ± 0.019 s	420.748 ± 0.110 j
S34	0.140 ± 0.013 fghijkl	4.925 ± 0.027 mn	0.463 ± 0.017 w	0.032 ± 0.002 lm	0.083 ± 0.003 i	5.842 ± 0.009 q	456.249 ± 0.090 i
S35	0.142 ± 0.010 defgh	4.560 ± 0.013 q	0.669 ± 0.017 u	0.021 ± 0.002 op	0.044 ± 0.003 m	4.932 ± 0.017 v	352.468 ± 0.095 l
CV(%)	51.7	31.95	62.09	43.17	131.07	32.11	79.72

Means with different letters in the same column express significant differences (Duncan’s test *p* < 0.05).

**Table 6 foods-12-03615-t006:** Identification of volatile compounds in fruits of different *A. arguta* resources using HS-GC-IMS.

Number	Count	Compound	CAS#	Formula	MW	RI	Rt [sec]	Dt [a.u.]	Comment
1	Esters	Methyl butanoate M	623-42-7	C5H10O2	102.1	1018.9	306.187	1.14902	Monomer
2		Methyl butanoate D	623-42-7	C5H10O2	102.1	1010.8	300.593	1.43148	Dimer
3		Methyl acetate	79-20-9	C3H6O2	74.1	890	242.237	1.19625	
4		Isoamyl acetate M	123-92-2	C7H14O2	130.2	1146.5	422.748	1.31005	Monomer
5		Isoamyl acetate D	123-92-2	C7H14O2	130.2	1141.9	417.108	1.75368	Dimer
6		Hexyl propanoate	2445-76-3	C9H18O2	158.2	1300.6	663.746	1.42868	
7		Hexyl acetate	142-92-7	C8H16O2	144.2	1298.1	660.409	1.38933	
8		Ethyl (E)-hex-2-enoate	27829-72-7	C8H14O2	142.2	1044.1	324.256	1.31395	
9		Ethyl propionate M	105-37-3	C5H10O2	102.1	966.4	276.19	1.14517	Monomer
10		Ethyl propionate D	105-37-3	C5H10O2	102.1	984.3	284.817	1.45669	Dimer
11		Ethyl hexanoate M	123-66-0	C8H16O2	144.2	1256.9	585.898	1.34038	Monomer
12		Ethyl hexanoate D	123-66-0	C8H16O2	144.2	1248.9	571.997	1.80357	Dimer
13		Ethyl formate	109-94-4	C3H6O2	74.1	854.4	227.914	1.0705	
14		Ethyl butyrate	105-54-4	C6H12O2	116.2	1053.1	331.029	1.55657	
15		Ethyl acetate M	141-78-6	C4H8O2	88.1	919.2	254.721	1.10585	Monomer
16		Ethyl acetate D	141-78-6	C4H8O2	88.1	918	254.194	1.33838	Dimer
17		Ethyl isovalerate	108-64-5	C7H14O2	130.2	1077	349.558	1.65689	
18		Butyl propionate	590-01-2	C7H14O2	130.2	1174.4	458.567	1.71886	
19		Butyl acetate M	123-86-4	C6H12O2	116.2	1034.3	317.103	1.23496	Monomer
20		Butyl acetate D	123-86-4	C6H12O2	116.2	1035.3	317.832	1.61627	Dimer
21		Butyl acrylate	141-32-2	C7H12O2	128.2	887	240.999	1.26357	
22		Butyl isovalerate	109-19-3	C9H18O2	158.2	1011.2	300.863	1.3947	
23		1-Methoxy-2-propyl acetate	108-65-6	C6H12O3	132.2	857.5	229.122	1.14191	
24		Citronellyl formate	105-85-1	C11H20O2	184.3	1288.5	643.76	1.8982	
25	Alcohols	Ethanol	64-17-5	C2H6O	46.1	984.1	284.691	1.04754	
26		Cis-2-Penten-1-ol	1576-95-0	C5H10O	86.1	1342.4	721.899	0.94816	
27		1-Penten-3-ol	616-25-1	C5H10O	86.1	1176.3	461.09	0.94578	
28		Isobutanol	78-83-1	C4H10O	74.1	1149.3	426.234	1.36406	
29		Carveol M	99-48-9	C10H16O	152.2	1242.2	560.754	1.29522	Monomer
30		Carveol D	99-48-9	C10H16O	152.2	1237.4	552.68	1.68177	Dimer
31		3-Methyl-1-butanol	123-51-3	C5H12O	88.1	1223.3	529.961	1.49475	
32		1-Butanol	71-36-3	C4H10O	74.1	1160.7	440.596	1.18265	
33		Cyclooctanol	696-71-9	C8H16O	128.2	1164.6	445.668	1.12941	
34		2-Methyl-1-butanol	137-32-6	C5H12O	88.1	1180.1	466.173	1.47668	
35		1-Pentanol	71-41-0	C5H12O	88.1	1272.9	614.561	1.25548	
36		1-Hexanol M	111-27-3	C6H14O	102.2	1375.3	771.107	1.32787	Monomer
37		1-Hexanol D	111-27-3	C6H14O	102.2	1373	767.501	1.64025	Dimer
38		1-Hexanol T	111-27-3	C6H14O	102.2	1367.9	759.689	1.98315	Trimer
39		Cineole M	470-82-6	C10H18O	154.3	1216.4	519.23	1.29225	Monomer
40		Cineole D	470-82-6	C10H18O	154.3	1216.7	519.575	1.72287	Dimer
41		Leaf alcohol	928-96-1	C6H12O	100.2	1383.9	784.497	1.23283	
42		2-Heptanol	543-49-7	C7H16O	116.2	1292.5	651.413	1.71865	
43	Ketones	2-Octanone	111-13-7	C8H16O	128.2	1304.1	668.411	1.33533	
44		L(-)-Carvone	6485-40-1	C10H14O	150.2	1137	411.188	1.81159	
45		Isomenthone	491-07-6	C10H18O	154.3	1178.9	464.569	1.34028	
46		2-Hexanone	591-78-6	C6H12O	100.2	1064.4	339.629	1.50148	
47		2-Heptanone M	110-43-0	C7H14O	114.2	1194.2	485.826	1.25783	Monomer
48		2-Heptanone D	110-43-0	C7H14O	114.2	1201.1	495.975	1.63226	Dimer
49		Cyclohexanone	108-94-1	C6H10O	98.1	1300.3	663.412	1.15313	
50		2-Butanone M	78-93-3	C4H8O	72.1	894.9	244.296	1.06226	Monomer
51		2-Butanone D	78-93-3	C4H8O	72.1	937.1	262.631	1.2478	Dimer
52		5-Methyl-3-heptanone M	541-85-5	C8H16O	128.2	942.3	265.002	1.27861	Monomer
53		5-Methyl-3-heptanone D	541-85-5	C8H16O	128.2	961.6	273.911	1.68433	Dimer
54		Methyl isobutenyl ketone	141-79-7	C6H10O	98.1	1155.1	433.411	1.44875	
55		3-Hydroxy-2-butanone	513-86-0	C4H8O2	88.1	1307.8	673.432	1.05977	
56		3-Hepten-2-one	1119-44-4	C7H12O	112.2	932.2	260.463	1.2265	
57		3,5-Dimethyl-1,2-cyclopentanedione	13494-07-0	C7H10O2	126.2	1066.3	341.109	1.61079	
58		3,4-Dimethyl-1,2-cyclopentanedione	13494-06-9	C7H10O2	126.2	1093.2	362.744	1.62262	
59		1-Penten-3-one	1629-58-9	C5H8O	84.1	1058.9	335.428	1.0793	
60		Hydroxyacetone	116-09-6	C3H6O2	74.1	1277.9	623.753	1.04359	
61		2-Pentanone	107-87-9	C5H10O	86.1	951.4	269.186	1.37493	
62	Aldehydes	Hexanal M	66-25-1	C6H12O	100.2	1118.7	389.828	1.25902	Monomer
63		Hexanal D	66-25-1	C6H12O	100.2	1094.5	363.792	1.56769	Dimer
64		Heptanal M	111-71-7	C7H14O	114.2	1202.9	498.672	1.33033	Monomer
65		Heptanal D	111-71-7	C7H14O	114.2	1202.9	498.672	1.69473	Dimer
66		Butanal M	123-72-8	C4H8O	72.1	878.1	237.336	1.11738	Monomer
67		Butanal D	123-72-8	C4H8O	72.1	867	232.889	1.2832	Dimer
68		Benzaldehyde	100-52-7	C7H6O	106.1	1531.1	1053.979	1.15444	
69		Isovaleraldehyde	590-86-3	C5H10O	86.1	938.6	263.336	1.40951	
70		trans-2-Pentenal	1576-87-0	C5H8O	84.1	1150	427.068	1.10704	
71		2-Methylbutyraldehyde	96-17-3	C5H10O	86.1	875.4	236.261	1.1511	
72		Isobutyraldehyde M	78-84-2	C4H8O	72.1	817.6	213.951	1.09932	Monomer
73		Isobutyraldehyde D	78-84-2	C4H8O	72.1	852.8	227.247	1.28367	Dimer
74		(Z)-4-Heptenal	6728-31-0	C7H12O	112.2	1300.2	663.227	1.61962	
75		trans-2-Pentenal	1576-87-0	C5H8O	84.1	1112	382.209	1.36162	
76		trans-2-Hexena M	6728-26-3	C6H10O	98.1	1251.7	576.747	1.1827	Monomer
77		trans-2-Hexenal D	6728-26-3	C6H10O	98.1	1224.3	531.583	1.51357	Dimer
78		Propionaldehyde	123-38-6	C3H6O	58.1	826.2	217.111	1.04325	
79	Terpenes	Dipentene M	138-86-3	C10H16	136.2	1210.7	510.409	1.21981	Monomer
80		Dipentene D	138-86-3	C10H16	136.2	1215.6	517.85	1.72287	Dimer
81		Camphene	79-92-5	C10H16	136.2	1080.1	352.008	1.20989	
82		β-Pinene	127-91-3	C10H16	136.2	1134.7	408.475	1.21824	
83		Myrcene	123-35-3	C10H16	136.2	1190.1	480.08	1.21772	
84		alpha-Pinene	80-56-8	C10H16	136.2	1033.8	316.769	1.22179	
85		α-Phellandrene	99-83-2	C10H16	136.2	1174.6	458.757	1.21952	
86		Terpinolene	586-62-9	C10H16	136.2	1292.5	651.428	1.21948	
87	Acids	Acetic acid M	64-19-7	C2H4O2	60.1	1504.8	999.756	1.05441	Monomer
88		Acetic acid D	64-19-7	C2H4O2	60.1	1505	1000.243	1.15277	Dimer
89		Isovaleric acid	503-74-2	C5H10O2	102.1	863.4	231.439	1.21454	
90	Pyrazines	2-Methoxy-3-methylpyrazine	2847-30-5	C6H8N2O	124.1	985	285.138	1.57071	
91		2,3,5-Trimethylpyrazine	14667-55-1	C7H10N2	122.2	1445.7	887.951	1.17114	
92		2-Ethyl-3-methylpyrazine	15707-23-0	C7H10N2	122.2	1337.9	715.399	1.59816	
93	Furans	2,5-Dimethylfuran	625-86-5	C6H8O	96.1	930.2	259.546	1.02742	
94		2-Pentylfuran	3777-69-3	C9H14O	138.2	1228.3	537.902	1.24624	
95	Other compounds	Toluene	108-88-3	C7H8	92.1	1033.1	316.26	1.02501	
96		2,4,6-Collidine	108-75-8	C8H11N	121.2	1374.1	769.181	1.5841	

## Data Availability

All related data and methods are presented in this paper. Additional inquiries should be addressed to the corresponding author.

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
