# Peer review of "Characterization of Key Compounds of Organic Acids and Aroma Volatiles in Fruits of Different *Actinidia argute* Resources Based on High-Performance Liquid Chromatography (HPLC) and Headspace Gas Chromatography–Ion Mobility Spectrometry (HS-GC-IMS)"

_foods, 2023, doi:10.3390/foods12193615_

Round 1

Reviewer 1 Report

The manscript is too lengthy and should check wether the word count limits are within the journal set limits.

The language of the paper needs to be improved significantly.

In abstract, the acronyms like VIP needs to be elaborated when used for the first time. 

In Introduction part, Lane 10: 'and laxative' was repeated twice.

In page 2: first sentence is not clear and needs reframing

Figure 1 is not cited in the text and needs appropriate citation 

Figure 2 & 3 are not clear

Data in table 7 and 8 are exhastive and authors should see the feasibility of them being provided as supplementary files instead of listing directly in the main manuscript.

Conclusion section is too dsescriptive and repititive of the whole manuscript, which needs to be synopsized into a smaller paragraph indicating the final result of the current study. 

Needs significant improvement. Some are listed in the comments and may make use of professional language editing services, if required.

Reviewer 2 Report

This is a good paper looking at volatiles in A. arguta. The biochemical analyses are novel and the information is a good contribution to new knowledge in order to select new types of A. arguta. I consider that Tables 7 and 8 are too big and the information must be summarized.

Abstract

Methyl-1-butanol, 1-Hexanol or methyl o hexanol with small letters, I think here is not correct to use capital letters.

Introduction

Korean Peninsula, and the Russian Far East[2], etc. Please delete ,etc.

the consumption of models is too much[22]. It is not clear the meaning of too much in this sentence.

in kiwifruit[21], jujube[30], melons[31], wines[32], eggs[33], and honey[34], etc. Please delete etc.

In addition, based on volatile aroma compounds through multivariate statistical analysis, performed quantitative descriptive analysis of the data to analyze the differences in volatile aroma compounds between individual resources. Meanwhile, combined principal component analysis, OPLS-DA analysis and OAV analysis to screen essential volatile compounds affecting the fruit flavor of Actinidia arguta resources. Please improve the redaction. This paragraph is not clear to understand.

Please indicate clearly which is the objective of the study.

After use the words Actinidia arguta is recommended to use afterwards only A. arguta in all the text

Material and methods

The 35 Actinidia arguta resources for testing (see Table 1). Delete see

at the Institute of Specialty Products, Chinese Academy of Agricultural Sciences, and the fruit trees. I think A. arguta is not a fruit tree it is a vine. Thus, it is better to say and the vines with good……..

Guidelines for Postharvest Physiology and Biochemistry Experiments on Fruits and Vegetables. Please put a cite here

Results and analysis

acidity is too large, not easy to be accepted; but the sugar content is high, the edge is too low, the flavor is single and light, lack of sweet and sour moderate taste; sugar-acid are too low, even if there is a suitable sugar-acid ratio, but also people will feel bland and tasteless[42]. Improve the redaction of this paragraph.

In contrast, S12 has a higher sugar-acid ratio with titratable acid content at higher sugar content; therefore, its fruit flavor can be superior to its source. Please improve the redaction of this sentence.

The highest oxalic acid content was 0.182 g/L for S12, and the lowest was 0.013 g/L for S31. Please discuss this in terms of human health. The concentrations of oxalic acid found in the flesh may negatively affect human health ?.

The results of hierarchical clustering analysis(HCA) can better respond to the characteristics of organic acid substances in the fruit samples of different Actinidia arguta resources; according to the organic acid cluster analysis of each resource, it can be seen that when the value of the transverse tangent line is taken between 200 400, the 35 Actinidia arguta resource fruit samples are divided into six classes, the first class is S5 and S25, the second class is S4, S10 and S21, the third category is S27, S9 and S20, the fourth category………….

This could be table 5 instead of showing all the values, thus, you can delete them as well figure 1. Table 5 could show the groups.

Figures 2 and 3 are not clear, please improve it.

16 ketones, 12 aldehydes, 7 terpenoids, 3 pyrazines, 2 furans, 2 acids and 2 other compounds, which essentially cover the range of aroma compounds found in fruits[21, 52, 53]. I suggest including an additional cite here: Antioxidants in processed fruit, essential oil, and seed oils of feijoa. Guerra-Ramirez D et al. (2021). Not Bot Horti Agrobo 49(1):11988

Tables 7 and 8 are very big. You already used principal component and cluster analysis and then summarized this valuable information. Thus, you can mention that these tables are available in the Data Availability Statement. 

The edition improvement of some sentences and paragraphs is recommended.

Reviewer 3 Report

I have reviewed the manuscript entitled “Characterization of the Key Aroma Volatile Compounds in Different Actinidia argute Resources by Headspace Gas Chromatography-Ion Mobility Spectrometry (HS-GC-IMS), Orthogonal Partial Least Squares Discriminant Analysis (OPLS-DA) and Odor Activity Value (OAV)”. This work is well-presented. Experiments are well planned and the analyses were affected by appropriate methods.  There is sufficient discussion of the results obtained.

Detailed remarks about the text:

The names of the compounds in the abstract should be controlled and those in the text should be given in lower case.

Keywords: Please exclude words that are already included in the title.

The title is too long. It should be shorter and more understandable.

The process of sampling needs to be described in detail. It's important to acknowledge that the outcomes may be influenced by variations in agroecological conditions, annual climatic conditions in specific locations, cultural practices, and the number of samples subjected to analysis. Explain how homogeneity and traceability were ensured, and provide information about the number of replicates taken and analyzed.

2.3.1 Determination of soluble sugar and titratable acid content: The reference and year should be indicated.

"C18-XT (4.6*250*5) column" should be corrected to "C18-XT (4.6x250x5) column".

Table 2. Concentration ranges of the standards used should be indicated.

Is Table 7 formed by reference to a single internal standard? Which compound was used as the internal standard? What is the recovery yield of the internal standard? It has been assumed that the GC (FID) responses for the variety of volatile components present are the same as for the internal standard. This is not so, as GC (FID) responses vary widely depending on the compounds.

Please provide more detailed information about the FlavourSpec® flavour analyzer.

No methodology on OAV analysis is given in the materials and methods section.

The compound names given in Table 6 should be checked and corrected.

Figures 5-10: The resolutions are too low and not understandable.

Results and Discussion: While there are many tables present, the results have not been discussed in detail. Ensure that each table is appropriately interpreted and discussed in the context of the research objectives. This will help readers better understand the significance of the findings.

The manuscript needs to be improved in terms of language and presentation.
